# KDPRA: A Dual-Molecule Knowledge Distillation model with Cross-Attention Fusion for Protein–RNA Binding Affinity Prediction

## Abstract

Quantifying the binding affinity between proteins and RNAs is critical for understanding the recognition mechanisms underlying protein–RNA interactions. However, current computational methods face two major limitations: (1) the scarcity of training data, as experimentally measured protein–RNA binding affinity datasets are limited and insufficient to support the effective training of complex models; and (2) the lack of efficient cross-modal feature interaction mechanisms, which hampers the accurate modeling of the intricate binding patterns between proteins and RNAs. To tackle these challenges, we propose KDPRA, a protein–RNA binding affinity prediction model based on knowledge distillation and a cross-attention mechanism. To better learn residue-level representations of proteins and RNAs, we independently train teacher models for each modality and employ knowledge distillation to guide the student model to learn effective structural and semantic representations. Furthermore, KDPRA incorporates a bidirectional cross-attention fusion module to capture general patterns of protein–RNA interactions. Experimental results demonstrate that KDPRA outperforms existing methods. Case studies further reveal that KDPRA can effectively predict protein–RNA binding affinities, providing strong biological interpretability and promising application potential.

## 1 Introduction

Protein–RNA interactions play a crucial role in various biological processes, such as gene expression regulation (Keene, 2007), post-transcriptional regulation (Batista & Chang, 2013), and protein synthesis (Cirillo et al., 2013). These interactions are not only essential for normal cellular functions but also play a decisive role in numerous pathological processes (Batista & Chang, 2013), including certain neurodegenerative diseases (Cirillo et al., 2013). The binding affinity largely determines the functional outcome of protein–RNA complexes (Crocker et al., 2016), therefore, accurately predicting protein–RNA binding affinity is of great biological significance. It holds potential application value for understanding molecular mechanisms and developing RNA-targeted therapeutic strategies. Currently, methods for measuring the binding affinity of protein–RNA complexes primarily include isothermal titration calorimetry (ITC), surface plasmon resonance (SPR), electrophoretic mobility shift assay (EMSA), filter binding assay (FBA), and dynamic light scattering (DLS). Although these traditional experimental approaches can effectively measure binding affinity, they also suffer from limitations such as high cost, complex operation, and long measurement time.

In recent years, research on predicting the binding affinity of protein–RNA complexes using computational approaches has seen continuous development. Yang et al. (Yang et al., 2014) developed a template-based method, which utilizes a non-redundant structural template library of protein–RNA complexes. This approach employs fold recognition techniques to match the query protein sequence with protein structures in the template library, and then predicts the binding affinity between the query sequence and the template RNA based on significant matches. Nithin et al. (Nithin et al., 2019) proposed a method for predicting protein–RNA binding affinity by computing structural and physicochemical parameters of the protein–RNA interface. Deng et al. (Deng et al., 2019) developed a machine learning-based model called PredPRBA, which uses Gradient Boosting Regression Trees (GBRT) to predict protein–RNA binding affinity. In this method, protein–RNA complexes are

divided into six categories based on the RNA type, and an independent GBRT model is trained for each category. To incorporate more detailed structural information, Hong et al. (Hong et al., 2023a) conducted an in-depth characterization of the structures of protein–RNA complexes and applied least squares regression to predict their binding affinity.

In this paper, we propose KDPRA, a novel model for protein–RNA binding affinity prediction that integrates dual-teacher knowledge distillation and a bidirectional cross-attention mechanism. As illustrated in Figure 1, the overall framework of KDPRA consists of the following components: For the input protein and RNA, the model first extracts multi-source features from the protein, including sequence representations derived from the pretrained language model Evolutionary Scale Modeling v2 (ESM2), structural features from Dictionary of Secondary Structure of Proteins (DSSP), and a graph-level virtual node feature constructed based on structural hotspot regions (i.e., O-ring regions). The RNA representation is obtained from embeddings extracted by the pretrained RNA language model RNA-FM and structurally encoded via a Graph Attention Network (GAT) to yield residue-level embeddings. To address the scarcity of binding affinity data for protein–RNA complexes, we construct a dual-teacher distillation framework to enhance the model's representational capacity. Specifically, we pretrain a protein teacher model on a protein–protein affinity prediction task and an RNA teacher model on an RNA–small molecule affinity task. During training, the parameters of both teacher models are frozen, and feature-level distillation losses are applied to guide the protein and RNA student models toward joint learning of structural and semantic information.To further improve generalization, we introduce a residue-level RNA data augmentation strategy that includes embedding perturbation, fragment shuffling, and random residue masking. On top of this, we design a motif probing module that identifies key residues by sliding perturbation fragments and analyzing output variations, thereby indirectly capturing potential RNA-binding motifs or functional footprints. To model residue-level interactions between proteins and RNA more explicitly, we propose a bidirectional cross-attention fusion module, which captures latent binding patterns and interaction regions. The fused interaction representation is then fed into a regression head to predict the binding affinity of the protein–RNA complex. Our main contributions are summarized as follows:

- We design a dual-teacher knowledge distillation module that leverages prior knowledge from protein–protein and RNA–small molecule affinity prediction tasks. By performing feature-level distillation, the student model is enhanced in both structural and semantic modeling, leading to improved representation of protein–RNA complexes.

- We propose a bidirectional cross-attention module that captures residue-level interactions between proteins and RNA, effectively aligning cross-modal features and enhancing the aggregation and expression of information in key interaction regions.

- We introduce protein structural hotspot regions (O-ring) as prior structural knowledge and construct graph-level virtual node features, which provide high signal-to-noise context for protein structural modeling and improve the model's ability to perceive functionally critical residues.

- We develop a residue masking augmentation strategy and a motif probing module, which identifies critical RNA regions for binding by applying sliding perturbation fragments and analyzing changes in model outputs—thereby capturing potential RNA-binding motifs or functional footprints.

## 2 RELATED WORK

### 2.1 PROTEIN AND RNA LANGUAGE MODELS

Large language model (LLM) technologies have been progressively adapted to biological sequence prediction tasks. In the domain of Protein Language Models (PLMs), ESM-1b (Rives et al., 2021) pioneered the construction of a large-scale context-aware model trained on 250 million protein sequences (encompassing 86 billion amino acids), demonstrating the effectiveness of the unsupervised pretraining–fine-tuning paradigm for biological sequence modeling. Since then, PLMs have continued to evolve, with models such as ESM-1v (Meier et al., 2021), ESM-2 (Lin et al., 2023), and XTrimoPGLM (Chen et al., 2024a). The recently released ESM-3 (Hayes et al., 2025) introduces a

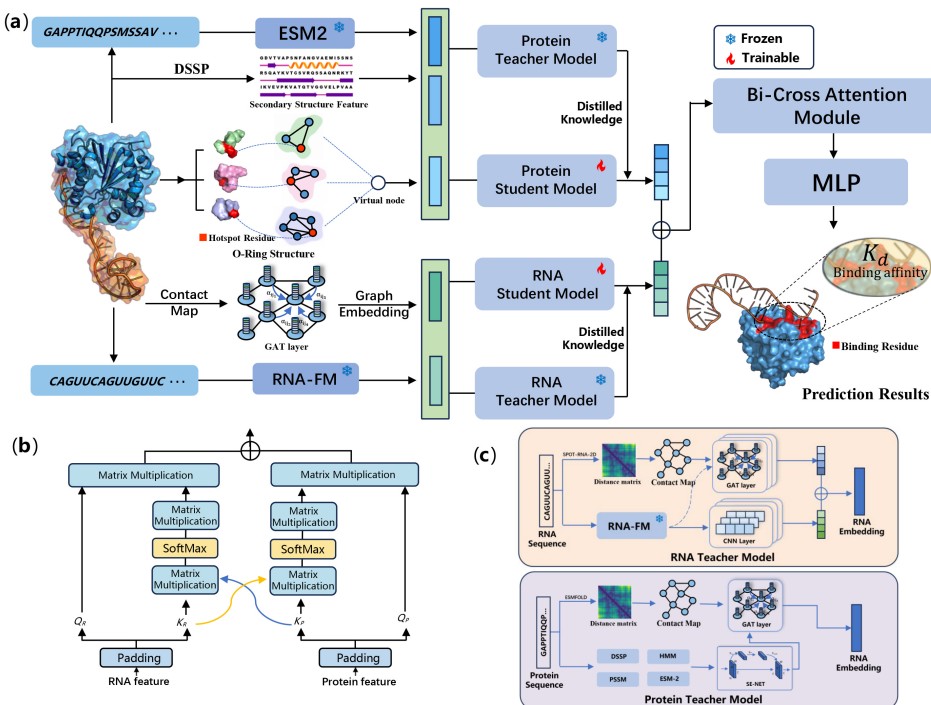

Figure 1: Workflow of KDPRA. (a) Overall framework. KDPRA extracts multi-source protein features from ESM2 embeddings, DSSP secondary structures, and hotspot-based virtual nodes. RNA embeddings are obtained from RNA-FM and structurally encoded with a GAT layer. A dual-teacher distillation strategy transfers knowledge from protein–protein and RNA–small molecule affinity models to protein and RNA student models. Residue-level embeddings are fused via a bidirectional cross-attention module and fed into a regression head to predict binding affinity. (b) Bidirectional cross-attention. Protein-to-RNA and RNA-to-protein attention branches capture residue-level cross-modal interactions through matrix multiplications and softmax-based attention scoring. (c) Teacher models. The protein teacher integrates ESM2, DSSP, HMM and PSSM features, while the RNA teacher combines RNA-FM embeddings with GAT-based structural encoding. Both teachers provide feature-level supervision for knowledge distillation.

sequence-structure co-generative framework that significantly enhances de novo design capabilities for functional proteins and enables cross-species generalization. Simultaneously, RNA Language Models (RLMs), closely linked to gene expression regulation, have focused on uncovering the functions of non-coding RNAs. Representative single-modality models include RNA-FM (Chen et al., 2022) and RNAErnie (Wang et al., 2024). On the cross-modal level, RhoFold+ (Shen et al., 2024) integrates RNA sequence features with protein structural information to predict RNA–protein interaction interfaces. Notably, AlphaFold3 (Abramson et al., 2024), a state-of-the-art multimodal framework, extends modeling capabilities to high-precision, dynamic structure prediction for protein–ligand and protein–RNA complexes through diffusion-driven conformational sampling and multi-component coupling modules, substantially reducing interface prediction errors in molecular interactions. Despite the powerful performance of PLMs and RLMs in single-molecule tasks, their cross-modal integration for applications such as gene regulation and synthetic biology remains in its early stages. There is an urgent need to develop unified multimodal architectures to unlock comprehensive modeling of complex biological systems.

## 2.2 KNOWLEDGE DISTILLATION

Knowledge distillation has demonstrated significant potential in bioinformatics by improving data processing efficiency and model performance, particularly when dealing with high-dimensional and complex biological data. Wang et al. (Wang et al., 2022) proposed trRosettaX-Single, an au-

tomated single-sequence-based protein structure prediction method, which leverages a supervised Transformer-based protein language model to extract sequence embeddings and applies knowledge distillation to optimize a multi-scale network architecture for accurate prediction of residue-level spatial relationships, ultimately reconstructing 3D structures through energy minimization. Zhao et al. (Zhao et al., 2024) developed a hybrid model combining an improved temporal convolutional network (TCN), bidirectional long short-term memory (BiLSTM), and multi-head attention (MHA) for 8-state and 3-state protein secondary structure prediction. Their method integrates a ProtT5-driven knowledge distillation strategy to significantly enhance generalization. Chen et al. (Chen et al., 2024b) introduced SEKD-PPIS, a novel deep learning framework combining equivariant graph neural networks (GNN) and self-distillation for protein–protein interaction site prediction. The model employs transfer learning from pretrained protein language models to capture deep semantic features and uses residual connections to mitigate gradient vanishing and over-smoothing, thereby improving representation learning and cross-scenario generalizability. Lu et al. (Lu et al., 2023) proposed KIDA, a knowledge distillation-driven model for drug–target affinity (DTA) prediction. KIDA distills interaction information from 3D drug–target complexes into a lightweight model, enabling accurate predictions without requiring structural docking. It uses 3D protein pocket structures and 2D molecular graphs as input, enhancing interpretability of binding mechanisms. To address challenges such as information loss and large model size caused by complex encoder networks, Yuan et al. (Yuan et al., 2022) developed FusionDTA, a deep learning framework for DTI prediction based on a novel multi-head linear attention mechanism. The framework dynamically integrates global contextual features and introduces a teacher–student distillation paradigm to reduce model complexity while maintaining predictive accuracy. Lastly, Geffen et al. (Geffen et al., 2022) designed DistilProtBert, a lightweight protein language model distilled from the large-scale ProtBert model. By extracting essential features via knowledge distillation, DistilProtBert preserves representation quality while reducing computational cost, making it suitable for resource-constrained protein analysis tasks.

## 3 MATERIALS

### 3.1 DATASETS FOR PROTEIN–RNA BINDING AFFINITY PREDICTION

The dataset used for model training is the **PRA_201** dataset constructed by Han et al (Han et al., 2025). It integrates samples from three public datasets: **PDBbind** (Wang et al., 2005), **PRBABv2** (Hong et al., 2023b), and **ProNAB** (Harini et al., 2022), comprising a total of 201 protein–RNA complexes. Each complex contains a single protein chain and a single RNA chain, and meets the following criteria: total protein residue length $L_p \leq 1000$, and total RNA base length $5 \leq L_r \leq 500$.

### 3.2 PRETRAINING DATASET FOR PROTEIN TEACHER MODEL

To train the protein teacher model, we utilized the dataset constructed by Nikam et al(Nikam et al., 2023). This dataset is based on the PDBBind v2020 database and was processed using the PISCES method (Wang & Dunbrack Jr, 2003) to remove redundancy by excluding samples with sequence similarity greater than 25%. This approach enhances the model's generalization capability across diverse protein structures. The final dataset comprises 903 protein-protein complexes, each accompanied by experimentally determined binding affinity (Kd) values. It encompasses six functional categories of complexes: antigen-antibody, enzyme-inhibitor, G protein-containing, receptor-containing, other enzymes, and miscellaneous complexes.

### 3.3 PRETRAINING DATASET FOR RNA TEACHER MODEL

The dataset used to train the RNA teacher model is derived from the R-SIM database(Krishnan et al., 2023). R-SIM catalogs experimentally validated interactions between RNA molecules and small compounds, comprising a total of 2,501 interaction records involving 461 distinct RNA targets and 1,288 unique small molecules. Following the preprocessing strategy adopted in the RSAPred method, we filtered and standardized the original R-SIM data, resulting in a curated dataset of 1,439 samples. These samples cover 341 RNA molecules and 749 small-molecule ligands. Each sample includes the RNA sequence, the SMILES representation of the small molecule, and the corresponding binding affinity value.

## 4 METHODS

### 4.1 OVERVIEW

We propose a protein–RNA binding affinity prediction model that integrates dual-teacher knowledge distillation and a bidirectional cross-attention mechanism. As illustrated in Figure 1, the model consists of four main components: a protein feature extraction module, an RNA feature extraction module, a knowledge distillation module, and a binding affinity prediction module. For protein representation, we combine three types of features: pre-trained embeddings from the ESM2 protein language model, structural features derived from DSSP, and a graph-level virtual node representation constructed from hotspot regions (O-ring areas) within the protein structure. For RNA, we utilize embeddings generated by the RNA-FM language model and encode structural information using a GAT, producing residue-level RNA representations. To address the scarcity of experimentally measured protein–RNA binding data, we adopt a dual teacher cross-task distillation framework to improve the model's representational capacity. Specifically, we pretrain two modality-specific teacher models using related affinity prediction tasks: one on protein–protein interactions and the other on RNA–small molecule interactions. During training, the parameters of the teacher models are kept frozen, and their representations are used to guide the learning of the student models for both protein and RNA via feature-level knowledge distillation. Furthermore, we design a bidirectional cross-attention module to capture fine-grained interactions between protein and RNA residues. This module allows the model to learn potential binding patterns and interface relationships. The fused representations are then fed into a regression head to produce the final prediction of protein–RNA binding affinity.

### 4.2 BI-DIRECTIONAL CROSS-ATTENTION FUSION MODULE

To effectively capture and integrate complementary information between protein and RNA modalities, we propose a **bi-directional cross-attention fusion module**. Unlike traditional one-way attention mechanisms, this module enables residues in each modality to attend to the contextual features of the other, thereby enhancing alignment and cross-modal interaction.

Let $P \in \mathbb{R}^{n \times d}$ and $R \in \mathbb{R}^{m \times d}$ denote the protein and RNA feature matrices, respectively, where $n$ and $m$ are the sequence lengths, and $d$ is the embedding dimension. The fusion process consists of two symmetric attention flows:

**(1) Protein-to-RNA Attention:** Each protein residue queries the RNA representation to selectively aggregate informative RNA contexts. This is achieved via:

$$Q_p = PW_Q^p, \quad K_r = RW_K^p, \quad V_r = RW_V^p, \tag{1}$$

$$A_{p \leftarrow r} = \mathrm{softmax}\left(\frac{Q_p K_r^\top}{\sqrt{d}}\right) V_r, \tag{2}$$

where $W_Q^p, W_K^p \in \mathbb{R}^{d \times d}$ and $W_V^p \in \mathbb{R}^{d \times d_v}$ are learnable projection matrices.

**(2) RNA-to-Protein Attention:** Similarly, RNA nucleotides query the protein representation to retrieve relevant residue-level contexts:

$$Q_r = RW_Q^r, \quad K_p = PW_K^r, \quad V_p = PW_V^r, \tag{3}$$

$$A_{r \leftarrow p} = \mathrm{softmax}\left(\frac{Q_r K_p^\top}{\sqrt{d}}\right) V_p. \tag{4}$$

Each attention pathway allows one modality to dynamically integrate relevant contextual signals from the other. To construct a unified cross-modality representation, we aggregate the outputs from both directions. One simple yet effective strategy is to average the pooled outputs:

$$F = \frac{1}{2}\left(\mathrm{pool}(A_{p \leftarrow r}) + \mathrm{pool}(A_{r \leftarrow p})\right), \tag{5}$$

where $\mathrm{pool}(\cdot)$ denotes mean pooling over the sequence dimension. Alternatively, the two outputs can be concatenated and further fused via a learnable MLP.

In summary, the proposed bi-directional cross-attention mechanism enables deep, fine-grained interaction between protein and RNA representations, effectively capturing mutual dependencies that are crucial for downstream affinity prediction.

### 4.3 CROSS-ATTENTION FUSION MODULE

The cross-attention mechanism has been proven to effectively capture and integrate sequence-to-sequence complementary information to enhance contextual alignment in sequence-to-sequence, thereby motivating its application to protein–RNA feature fusion. We employ a cross-attention module that enables each protein residue to selectively attend to RNA contexts. Given protein feature matrix $P \in \mathbb{R}^{n \times d}$ and RNA feature matrix $R \in \mathbb{R}^{m \times d}$, we first calculate linear projections as follows

$$Q = P W_Q, \quad K = R W_K, \quad V = R W_V$$

where $W_Q \in \mathbb{R}^{d \times d}, W_K \in \mathbb{R}^{d \times d}$ and $W_V \in \mathbb{R}^{d \times d_v}$ are learnable parameters. We then form attention logits by the scaled dot-product $\frac{QK^\top}{\sqrt{d}}$ and apply softmax to produce normalized weights that highlight pertinent RNA contexts for each protein residue. The cross-attention output is then calculated as

$$\mathrm{Attention}(Q, K, V) = \mathrm{softmax}\!\left(\frac{QK^\top}{\sqrt{d}}\right) V$$

Finally, we obtain a matrix that integrates RNA information into protein embeddings, where the $1/\sqrt{d_k}$ term prevents gradient vanishing or explosion at large dimensions.

### 4.4 KNOWLEDGE DISTILLATION MODULE

A wide range of studies have demonstrated that knowledge distillation (KD) markedly enhances student model performance by transferring rich, intermediate feature representations from larger. We incorporate the pre-trained RNA and protein teacher models into more compact student networks, thereby achieving significant model compression without sacrificing accuracy and substantially improving inference efficiency.In addition to compressing single-modality networks, our KD scheme facilitates multimodal fusion by aligning student embeddings across RNA and protein modalities. The shared distillation loss enforces that student RNA and proteins occupy compatible embedding spaces, thereby improving the subsequent cross-attention fusion between modalities.

We employ separate pairs of teacher-student for the RNA and protein branches. Each teacher model is a deep, high-capacity network pretrained on large-scale structural and sequence data. The RNA teacher model integrates graph attention layers with a convolutional neural network to capture both structural and sequential features of RNA sequences.The protein teacher model combines graph-attention layers with Transformer encoders to extract long-range dependencies among amino acid residues. The two student models respectively reflect the topological structure of their corresponding teacher models but with reduced dimensionality and fewer layers - facilitating,deployment in resource-constrained scenarios - while still capturing essential sequence features.

During the specific training process,we use the cosine similarity as the alignment metric.Given a batch of teacher embeddings$T \in \mathbb{R}^{B \times d}$and corresponding student embeddings $S \in \mathbb{R}^{B \times d}$, the cosine similarity for sample $i$ is

$$\cos \theta_i = \frac{T_i \cdot S_i}{\|T_i\| \, \|S_i\|},$$

where the dot product $T_i \cdot S_i$ sums element-wise products and $\| \cdot \|$ denotes the Euclidean norm.

Then, to convert similarity into a loss, we define

$$\mathcal{L}_{\mathrm{KD}} = 1 - \frac{1}{B} \sum_{i=1}^{B} \cos \theta_i$$

so that perfect alignment $\cos \theta_i = 1$ yields zero loss, while misalignment incurs a proportional penalty which preserves directional consistency in the embedding space and serves as a smooth, scale-invariant alignment mechanism.

## 5 RESULTS

### 5.1 METRICS AND IMPLEMENTATION DETAILS

We evaluate our model using four metrics: root mean square error (RMSE), mean absolute error (MAE), Pearson correlation coefficient (PCC), and Spearman correlation coefficient (SCC). For protein feature representation, we employ sequence embeddings extracted from the ESM-3B pre-trained model (2560 dimensions), combined with DSSP secondary structure features (14 dimensions) and an additional virtual node feature (2574 dimensions). For RNA sequences, we utilize representations obtained from the RNA-FM pre-trained model (640 dimensions).All experiments are conducted on four NVIDIA RTX 4090 GPUs. We adopt the Adam optimizer with an initial learning rate of $5 \times 10^{-4}$, a batch size of 16, and 2500 training epochs, using a cosine annealing scheduler for learning rate adjustment. Graph structures are constructed based on pairwise residue distance matrices, where a distance cutoff of 10Å is applied to determine edge connections for both proteins and RNAs. Descriptions of baseline and comparison methods are provided in the Appendix.

### 5.2 PREDICTING PROTEIN-RNA BINDING AFFINITY

We evaluate the performance of our model on the PRA_201 dataset. As shown in Table 1, the KD-PRA model achieves 0.593, 0.656, 0.820, and 1.100 for SCC, PCC, MAE, and RMSE, respectively. In addition, we compare KDPRA with four representative baseline methods covering both sequence-based and structure-based approaches, including DeepNAP (Pandey et al., 2024), FoldX (Delgado et al., 2025), PredPRBA (Deng et al., 2019), and CoPRA (Han et al., 2025). Table 1 shows that KDPRA outperforms all competing methods across all evaluation metrics. Specifically, compared with the second-best model, KDPRA achieves relative improvements of 12.74% and 22.85% in SCC and PCC, respectively.

Table 1: Performance comparison on the PRA201 dataset.

| Method | LM | Seq | Struc | SCC↑ | PCC↑ | MAE↓ | RMSE↓ |
|--------|----|----|-------|------|------|------|-------|
| DeepNAP | – | ✓ | – | 0.349 | 0.345 | 1.600 | 1.964 |
| FoldX | – | – | ✓ | 0.268 | 0.212 | – | – |
| PredPRBA | – | – | ✓ | 0.316 | 0.370 | 1.695 | 2.238 |
| CoPRA | ✓ | ✓ | ✓ | 0.526 | 0.534 | 1.172 | 1.428 |
| **KDPRA** | ✓ | ✓ | ✓ | **0.593** | **0.656** | **0.820** | **1.100** |

### 5.3 MODULE ABLATION STUDY

In this section, we systematically evaluate the impact of each key module on the model's predictive performance. We design several model variants and conduct comparative analysis under a five-fold cross-validation (CV) setting. Specifically, w/o KD denotes the complete removal of the knowledge distillation module, while w/o bi-cross refers to the removal of the bidirectional cross-attention mechanism. In addition, we introduce two more fine-grained ablation settings: only RNA KD applies distillation solely to the RNA branch, leaving the protein branch undistilled; conversely, only Protein KD applies distillation only to the protein branch, without involving the RNA branch. Table 4 presents the performance of different module combinations across several evaluation metrics,

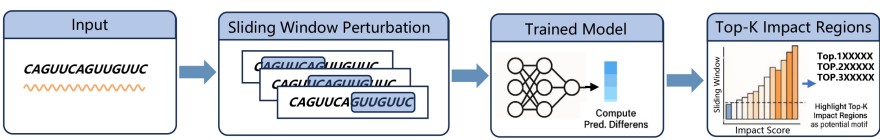

Figure 2: Workflow of the motif probing module. The module applies a sliding window perturbation strategy to RNA embeddings, passes perturbed inputs through the model, and identifies top-k fragments with highest impact scores as critical regions.

including PCC, RMSE, SCC, and MAE. The results show that the model achieves the best performance when all modules are integrated. Completely removing the knowledge distillation module (w/o KD) leads to a significant performance drop, indicating that the distillation mechanism plays a crucial role in enhancing feature representation. Furthermore, both only RNA KD and only Protein KD outperform the non-distilled baseline, but neither surpasses the dual-branch distillation strategy. This suggests that knowledge transfer from both modalities is complementary in modeling binding affinity. Although removing the bidirectional cross-attention module (w/o bi-cross) also results in performance degradation, the impact is relatively smaller compared to the removal of the distillation module. This trend is particularly evident in correlation-based metrics such as PCC and SCC, further highlighting the central role of knowledge distillation in improving model generalization and capturing critical interaction features.

Table 2: Performance comparison of different module settings under five-fold cross-validation.

| Model Setting | PCC ↑ | SCC ↑ | RMSE ↓ | MAE ↓ |
|---|---|---|---|---|
| Full Model | 0.6566 | 0.5935 | 1.1007 | 0.8205 |
| w/o Bi-Cross | 0.6252 | 0.5806 | 1.1023 | 0.8725 |
| w/o KD | 0.5500 | 0.5700 | 1.3720 | 1.0980 |
| only RNA KD | 0.6234 | 0.6182 | 1.1335 | 0.8506 |
| only Protein KD | 0.6360 | 0.5459 | 1.5913 | 1.3106 |

## 5.4 MOTIF PROBING VIA LOCAL PERTURBATION-BASED DATA AUGMENTATION

To mitigate the limited training data, we propose a local fragment shuffling strategy that perturbs the order of RNA residues to simulate biological variations and enhance model robustness. During experiments, we observed that perturbing certain fragments significantly degraded prediction performance, suggesting their functional relevance—similar to in vitro selection methods that identify high-affinity binding motifs via RBP preferences (Keene, 2007).To systematically assess the model's sensitivity to local RNA segments, we design a motif probing module based on a sliding window strategy. The overall workflow is illustrated in Figure 2. It shuffles consecutive segments in RNA embeddings and quantifies their influence on model output. In a case study on the 1MMS complex, the fragment "UCAC" had the most significant impact, indicating potential motif functionality. We further visualized the model's positional sensitivity using a one-dimensional heatmap, as illustrated in Figure 3, where "UCAC" exhibited the highest impact score. Motif validation via the ATtRACT database (Giudice et al., 2016) confirmed that "UCAC" corresponds to a known binding site of the Arabidopsis thaliana HEN1 protein, and also appears in the NOVA2 binding region in Mus musculus, supported by CLIP-seq evidence. These results highlight both the biological relevance and cross-species conservation of the motif, underscoring our method's ability to identify functionally significant RNA patterns.

## 5.5 FEATURE ABLATION STUDY

In this section, we evaluate the contribution of different input features to the model's performance by conducting a series of feature ablation experiments. In each experiment, one specific modality feature is removed to analyze its impact on predictive capability. Specifically, w/o DSSP indicates

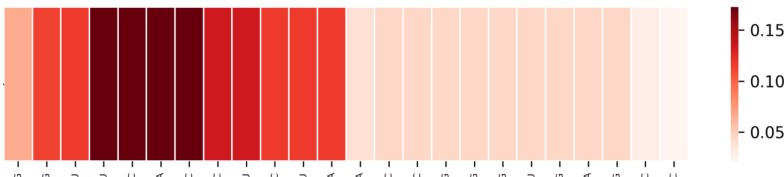

Figure 3: Impact heatmap of RNA residues in motif probing. The heatmap shows positional sensitivity of RNA residues, where the fragment "UCAC" exhibits the highest impact score. Database validation confirmed it as a conserved protein–RNA binding motif.

removing protein secondary structure features generated by DSSP; w/o ESM refers to the removal of protein sequence embeddings from the ESM model; w/o Virtual Node denotes eliminating the virtual node feature in the protein graph structure; and w/o RNA-FM represents excluding RNA sequence embeddings from the RNA Foundation Model. Table 3 summarizes the results in terms of PCC, SCC, RMSE, and MAE.

Experimental results show that the full model achieves the best overall performance with PCC = 0.6566, SCC = 0.5935, RMSE = 1.1007, and MAE = 0.8205. Among the ablation settings, removing RNA-FM features causes the largest degradation (RMSE = 1.7126, MAE = 1.3283), suggesting that RNA sequence representations play a more crucial role than previously assumed. Eliminating the virtual node feature also significantly reduces performance (PCC drops to 0.5840, MAE increases to 1.0404), highlighting its importance for capturing graph-level contextual information. Removing ESM and DSSP features leads to moderate performance decreases, with PCC reductions of 0.0370 and 0.0097 compared to the full model, respectively. This indicates that while pre-trained protein language models and secondary structure features provide complementary information, their absence is less detrimental than removing RNA-FM or virtual nodes.Overall, these results confirm that all modality features contribute to the final model performance, with RNA-FM and virtual node representations being particularly critical for accurate protein–RNA binding affinity prediction.

Table 3: Ablation study results on the PRA_201 dataset.

| Model Setting | PCC↑ | SCC↑ | RMSE↓ | MAE↓ |
|---|---|---|---|---|
| Full Model | 0.6566 | 0.5935 | 1.1007 | 0.8205 |
| w/o DSSP | 0.6469 | 0.6386 | 1.1170 | 0.8727 |
| w/o ESM | 0.6196 | 0.5947 | 1.1209 | 0.8806 |
| w/o Virtual Node | 0.5840 | 0.6053 | 1.3081 | 1.0404 |
| w/o RNA-FM | 0.6321 | 0.5012 | 1.7126 | 1.3283 |

## 6 CONCLUSION

In this paper, we propose KDPRA, a novel framework for protein–RNA binding affinity prediction that integrates dual-teacher knowledge distillation and a bidirectional cross-attention mechanism. To effectively represent protein and RNA, our model incorporates a multi-source feature extraction module, enabling comprehensive encoding of each component. To address the scarcity of protein–RNA complex data, we introduce two separately trained teacher models for protein and RNA, allowing the student model to inherit biologically meaningful knowledge through related pretraining tasks. During student model training, we further design a residue-level RNA data augmentation strategy to enrich the input space, along with a bidirectional cross-attention fusion module to explicitly model the interactions between protein and RNA. The resulting joint embeddings are then used to predict binding affinity. Extensive experiments, including five-fold cross-validation, ablation studies, and visualization analyses, consistently demonstrate the effectiveness and robustness of our proposed method.

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

# SUPPLEMENTARY MATERIAL

## A. STATEMENT

### 6.1 ETHICS STATEMENT

This work adheres to the ICLR Code of Ethics. In this study, no human subjects or animal experimentation was involved. All datasets used were sourced in compliance with relevant usage guidelines, ensuring no violation of privacy. We have taken care to avoid any biases or discriminatory outcomes in our research process. No personally identifiable information was used, and no experiments were conducted that could raise privacy or security concerns. We are committed to maintaining transparency and integrity throughout the research process.

### 6.2 REPRODUCIBILITY STATEMENT

We have made every effort to ensure that the results presented in this paper are reproducible. The experimental setup, including training steps, model configurations, and hardware details, is described in detail in the paper.

Additionally, all datasets used in this paper are publicly available resources(https://anonymous.4open.science/r/KDPRA-7B8E), ensuring the consistency and reproducibility of the evaluation results.

We believe these measures will enable other researchers to reproduce our work and further advance the field.

### 6.3 LLM USAGE

Large Language Models (LLMs) were used to aid in the writing and polishing of the manuscript. Specifically, we used an LLM to assist in refining the language, improving readability, and ensuring clarity in various sections of the paper. The model helped with tasks such as sentence rephrasing, grammar checking, and enhancing the overall flow of the text.

It is important to note that the LLM was not involved in the ideation, research methodology, or experimental design. All research concepts, ideas, and analyses were developed and conducted by the authors. The contributions of the LLM were solely focused on improving the linguistic quality of the paper, with no involvement in the scientific content or data analysis.

The authors take full responsibility for the content of the manuscript, including any text generated or polished by the LLM. We have ensured that the LLM-generated text adheres to ethical guidelines and does not contribute to plagiarism or scientific misconduct.

## B. FEATURE REPRESENTATIONS

The proposed method utilizes a combination of features for residue representation, including DSSP, pre-trained embeddings generated by ESM-2, and pre-trained embeddings generated by RN-FM.

### 6.3.1 DSSP

In our model, each amino acid residue is encoded with a 14-dimensional DSSP feature vector. This feature comprises an 8-dimensional one-hot representation of secondary structure types, distinguishing $\alpha$-helix, $\beta$-bridge, $\beta$-strand, 3-10 helix, $\pi$-helix, turn, bend, and coil. The remaining 6 dimensions describe solvent accessibility, backbone dihedral angles ($\phi$ and $\psi$), hydrogen bond counts, and other local geometric properties, thereby capturing both the fundamental secondary structure information and fine-grained spatial and chemical environment details.

### 6.3.2 PROTEIN LANGUAGE MODEL REPRESENTATION

ESM-2 (Evolutionary Scale Modeling 2) is a large-scale end-to-end protein language model that efficiently captures both sequence and structural features of proteins through deep learning methods. This representation not only includes the sequential information of proteins but also encodes the spatial interaction information among residues. In this study, we adopt a 3-billion-parameter sequence-based language model trained on the UniRef50 dataset (Suzek et al., 2007) and extract a 2560-dimensional sequence embedding for each residue.

### 6.3.3 RNA LANGUAGE MODEL REPRESENTATION

RNA-FM is a large-scale pre-trained language model specifically designed for RNA sequences. Built upon a Transformer architecture, RNA-FM is trained in a self-supervised manner on 23 million RNA transcripts, enabling it to capture contextual relationships between nucleotides as well as potential secondary structural information. Through masked language modeling, RNA-FM learns rich semantic features, including conserved sequence motifs, structural domains, and long-range dependencies.

### C. INPUT OF RELATED MODELS AND DEEPHOTRESI

We categorize the input features into five groups: (i) *sequence features* (e.g., position-specific scoring matrices, local structural entropy, conservation scores), (ii) *structure features* (e.g., secondary structure, energy scores), (iii) *solvent exposure features* (e.g., half-sphere exposure, residue depth, coordination number), (iv) *residue interaction network features* (e.g., betweenness centrality, closeness centrality, degree), and (v) *pre-trained embedding features* derived from protein language models.

Among existing methods, *DeePNAP* relies solely on sequence features without incorporating structural, solvent exposure, network-based, or pre-trained embedding information. *PredPRBA* utilizes traditional handcrafted features covering the first four categories but does not include pre-trained embeddings. *CoPRA* combines all five categories, including embeddings from pre-trained language models, while *FoldX* focuses on structure-, solvent-, and network-based energy features without sequence or embedding inputs.

Our proposed *KDPRA* integrates multi-source features, leveraging both handcrafted descriptors and pre-trained embeddings to achieve a more comprehensive residue representation. Details of the feature usage for each method are summarized in Table S1.

Table 4: Feature comparison among different protein-nucleic acid affinity prediction methods.

| Method | Sequence features | Structure features | Solvent exposure | Residue network | Pre-trained embedding |
|---|---|---|---|---|---|
| DeepNAP | ✓ | ✗ | ✗ | ✗ | ✗ |
| PredPRBA | ✓ | ✓ | ✓ | ✓ | ✗ |
| CoPRA | ✓ | ✓ | ✓ | ✓ | ✓ |
| FoldX | ✗ | ✓ | ✓ | ✓ | ✗ |
| KDPRA | ✓ | ✓ | ✓ | ✓ | ✓ |

### D. EVALUATION METRICS

To assess the regression performance of our proposed model, we adopt four widely used evaluation metrics: Spearman correlation coefficient (SCC), Pearson correlation coefficient (PCC), mean absolute error (MAE), and root mean squared error (RMSE). The definitions are given below:

1. **SCC**: Measures the rank correlation between the predicted and true binding affinity values, capturing monotonic relationships.

$$\text{SCC} = 1 - \frac{6 \sum_{i=1}^{n} d_i^2}{n(n^2 - 1)} \tag{6}$$

where $d_i$ is the difference between the ranks of predicted and actual values.

2. **PCC**: Evaluates the linear correlation between predictions and ground truth.

$$\text{PCC} = \frac{\sum_{i=1}^{n}(y_i - \bar{y})(\hat{y}_i - \bar{\hat{y}})}{\sqrt{\sum_{i=1}^{n}(y_i - \bar{y})^2}\sqrt{\sum_{i=1}^{n}(\hat{y}_i - \bar{\hat{y}})^2}} \tag{7}$$

where $y_i$ and $\hat{y}_i$ denote actual and predicted values, and $\bar{y}$ and $\bar{\hat{y}}$ are their respective means.

3. **MAE**: Represents the average magnitude of absolute prediction errors.

$$\text{MAE} = \frac{1}{n}\sum_{i=1}^{n}|y_i - \hat{y}_i| \tag{8}$$

4. **RMSE**: Quantifies the square root of the average squared differences between predictions and actual values.

$$\text{RMSE} = \sqrt{\frac{1}{n}\sum_{i=1}^{n}(y_i - \hat{y}_i)^2} \tag{9}$$

E. IMPACT OF CONTACT DISTANCE CUTOFF

To investigate the effect of the contact distance cutoff on constructing the protein residue contact graph and the subsequent binding affinity prediction, we conducted an ablation study with six different cutoffs: $2\,\text{Å}$, $4\,\text{Å}$, $6\,\text{Å}$, $8\,\text{Å}$, $10\,\text{Å}$, and $12\,\text{Å}$. In our graph construction, two residues are considered connected if the Euclidean distance between their $C_\alpha$ atoms is less than or equal to the selected cutoff. The resulting graphs were used as input to our proposed model while keeping all other hyperparameters unchanged. We evaluated the model performance for each cutoff setting using the regression metrics described in Section 6.3.3 (SCC, PCC, MAE, and RMSE). Among all tested cutoffs, a cutoff of $10\,\text{Å}$ achieved the lowest MAE (0.8205) and RMSE (1.1007), while also yielding a high PCC (0.6566). Based on this overall performance, we selected $10\,\text{Å}$ as the default contact distance cutoff in our final model. This finding indicates that moderately increasing the contact distance effectively captures critical residue-level interactions and improves binding affinity prediction accuracy.

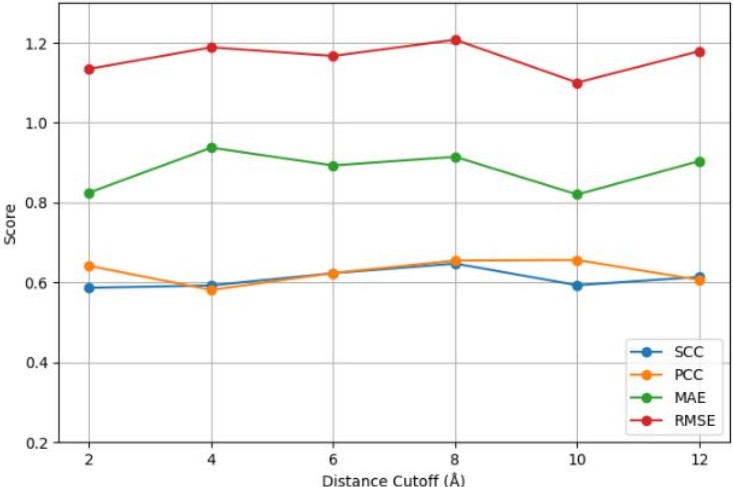

Figure S4: Results of different dataset sizes.

