# OpenReview forum: "KDPRA: A Dual-Molecule Knowledge Distillation model with Cross-Attention Fusion for Protein–RNA Binding Affinity Prediction"
_ICLR.cc/2026/Conference — ICLR 2026 Conference Withdrawn Submission_

### Official Review · Reviewer_jHrJ · 2025-10-26

**Soundness:** 2
**Presentation:** 3
**Contribution:** 2
**Rating:** 2
**Confidence:** 4

**Summary:**

This paper proposes KDPRA, protein–RNA binding-affinity predictor that tackles scarce training data and weak cross-modal interaction modeling by combining dual-teacher knowledge distillation with a bidirectional cross-attention fusion module. On the protein side, it fuses ESM2 sequence embeddings, DSSP secondary-structure features, and a graph-level “virtual node” built from structural hotspot (O-ring) regions. On the RNA side, it uses RNA-FM embeddings and encodes structure with a GAT to produce residue-level representations. Two modality-specific teacher models (one trained for protein–protein affinity and one for RNA–small-molecule affinity) are frozen and distill feature-level knowledge to lighter student branches for proteins and RNAs. A bidirectional cross-attention module explicitly models residue-to-residue interactions in both directions (protein-to-RNA and RNA-to-protein) before a regression head outputs affinity. The authors also run a comparison with a few other methods and perform ablation studies (across modules and features).

**Strengths:**

- The paper overall is well-presented and tackles an important problem in biology and therapeutics.
- It proposes a genuinely new architecture that combines dual-teacher knowledge distillation (protein–protein and RNA–small-molecule teachers) with bidirectional cross-attention to model residue-level protein-RNA interactions.
- It shows better performance on PRA201 with consistent gains across SCC, PCC, MAE, and RMSE, including sizable relative improvements in correlation vs. the next-best baseline.
- Careful module ablations that isolate what helps most (distillation is more important than bi-cross-attention), showing dual-branch distillation is complementary and central to generalization.
- Feature ablations that quantify contribution from each input; they reveal RNA-FM embeddings and the protein “virtual node” are particularly impactful.
- An interpretability angle via “motif probing”: sliding-window perturbations identify affinity-critical RNA fragments, with external database validation of HEN1 and NOVA2 inferred motifs.
- Figures are well made and make the approach easy to follow.

**Weaknesses:**

- My biggest concern is that the benchmark is way too small (only 201 complexes) and there is a real risk of overfitting, especially when cross-validation is used. On the same note, confidence intervals will be rather large and determining “better” and “worse” will be tricky; many of the entries in the ablation tables are probably within each other’s confidence intervals.
- The most important feature in the ablation study seems to be RNA-FM, which essentially encodes the RNA sequence. I suspect that this model is trivially just mapping protein sequences to their binding motifs (e.g., NOVA2 to “UCAC”) and scans the RNA sequence for the presence of such motifs. Meaning that it is not actually learning which residues on the protein are interacting which bases in the RNA sequence.
- The largest gain in modules comes from distillation, which brings with it the heavy reliance on cross-domain teachers. It could mean the student may inherit biases irrelevant (or even misaligned) for protein–RNA biophysics.
- Structure dependence may limit real-world applicability. The model leans on 3D-structure–derived graphs and a tuned inter-residue distance cutoff. That’s feasible for curated PDB complexes, but many targets lack high-quality structures (especially RNA), so performance in structure-sparse settings is unclear. (On that, can that structure be replaced by say AlphaFold predictions?)
- The “motif probing” analysis is compelling but demonstrated in limited cases, leaving open how robust or comprehensive the interpretability is across the dataset. Most proteins that bind to RNAs have known motifs and sequence specificities. This should be probed on a more comprehensive scale.
- We read that distillation is more important than bi-cross-attention, but we don’t know which direction of cross-attention is more important. Is protein helping RNA or is RNA helping protein?

**Questions:**

- Most proteins interact with RNA molecules in very specific and predictable manners; meaning proteins have strong sequence preferences, as evidenced by thousands of CLIP-seq, SELEX and protein binding microarray (PBM) datasets. Databases such as [https://cisbp-rna.ccbr.utoronto.ca/index.php] have been collecting and curating these sequence motifs across hundreds of species. One can speculate that such models would be much better teachers than a RNA-small molecule teacher. More importantly, should the model be trained only on very limited protein-affinity data, or be a multi-task model that can be trained on more data types, including the aforementioned data types, and only be fine-tuned on the limited affinity data.
- RNA-binding proteins usually are Lego-like and are built from known domains [https://www.sciencedirect.com/science/article/pii/S1097276520301593], and these domains have known sequence specificities. Therefore, for many such proteins we have a good sense of which protein residues are interacting to which RNA bases. There is no such analysis in this paper. There are some resources that can be helpful, for example, experimental data in [https://www.nature.com/articles/nbt.3128] can be used to verify such connections.
- Did you assess performance as a function of sequence/structure similarity to the training set (true OOD vs. near-IID)? On a similar note, what are the per-family performance breakdowns?
- Is Section 4.3 added by mistake? Section 4.2 covers cross-attention already.

---

### Official Review · Reviewer_wcw6 · 2025-10-29

**Soundness:** 3
**Presentation:** 2
**Contribution:** 1
**Rating:** 2
**Confidence:** 3

**Summary:**

The paper proposes KDPRA, a protein–RNA binding affinity predictor that addresses data scarcity and cross-modal fusion via dual-teacher knowledge distillation (protein–protein and RNA–small molecule teachers) and a bidirectional cross-attention module. Protein features combine ESM2 embeddings, DSSP, and a graph-level virtual node derived from O-ring hotspots; RNA uses RNA-FM embeddings and GAT-based structural encoding. On PRA201, KDPRA outperforms baselines (DeepNAP, FoldX, PredPRBA, CoPRA) across RMSE/MAE/PCC/SCC and shows that knowledge distillation and bi-directional cross-attention contribute substantially. The motif probing via local perturbations suggests biologically meaningful and conserved RNA motifs.

**Strengths:**

Focuses on binding affinity, a key determinant of protein–RNA complex function, with potential for therapeutic design. Dual-teacher distillation transfers priors from related tasks and aligns modalities; bidirectional cross-attention captures residue-level interactions. Combines strong PLMs (ESM2/RNA-FM), structural priors (DSSP), and graph virtual nodes from O-ring hotspots to improve SNR. Consistent improvements over both sequence- and structure-based baselines across multiple metrics; ablations isolate the contribution of each module. Motif probing yields conserved, database-validated motifs, supporting biological plausibility.

**Weaknesses:**

PRA201 is relatively small and curated; generalization beyond this set (e.g., to other PRNA benchmarks or independent test sets) is unclear. Teachers trained on protein–protein and RNA–small molecule tasks may introduce inductive biases not fully validated for protein–RNA  affinity. The advantage of bidirectional vs single cross-attention is shown, but variants (multi-head counts, stacking depth, pooling choices, MLP fusion) are not systematically explored. Lacks confidence intervals or significance testing for key comparisons; effect sizes vs variance are not reported.

**Questions:**

1. How were train/val/test splits constructed to avoid homologous leakage across protein or RNA sequences? Can you report performance on an independent holdout benchmark or stricter non-redundant splits (e.g., ≤30% identity)?
2. The bidirectional cross-attention averages pooled outputs; have you evaluated concatenation+MLP, gating, or co-attention transformers across multiple layers/heads? How sensitive are results to head count, depth, and pooling?
3. What is the training/inference time and memory footprint vs baselines? How sensitive are results to graph cutoff (10 Å), virtual node construction (O-ring detection), and hyperparameters (learning rate, batch size, augmentation strength)?

---

### Official Review · Reviewer_yvgX · 2025-10-31

**Soundness:** 3
**Presentation:** 2
**Contribution:** 2
**Rating:** 4
**Confidence:** 4

**Summary:**

This paper proposes a data-efficient approach, KDPRA, for predicting protein–RNA binding affinity. To overcome the scarcity of labeled protein–RNA complexes, the authors employ a dual-teacher knowledge distillation framework: one teacher model is trained on protein–protein interaction data and the other on RNA–small molecule binding data. The distilled representations are fused using a bi-directional cross-attention mechanism to capture complementary information from both molecular modalities. The method is theoretically sound and shows some empirical improvements over selected baselines.

**Strengths:**

•  The idea of leveraging related interaction tasks through dual knowledge distillation is sensible and well-motivated given the limited protein–RNA data.
•  The bi-directional cross-attention fusion design is clear and potentially generalizable to other multimodal molecular settings.
•  The authors include meaningful ablations (teacher branches, cross-attention), and the motif-probing case study offers valuable insights into how the model captures sequence motifs relevant to binding.

**Weaknesses:**

Limited evaluation scope. Results are reported only on one dataset (PRA201) and with a small set of baselines. In contrast, recent works such as CoPRA evaluate on multiple datasets and include stronger baselines. It is unclear why the authors selected only four and why they used the CoPRA (scratch) variant rather than the full model.
Data leakage and split policy unclear. PRA201 integrates PDBbind, PRBABv2, and ProNAB, but the paper never clarifies how train/test splits were done. Without homology-based splitting, cross-contamination between folds is possible. This is especially concerning given the small dataset size.
RNA structural graph construction not described. The paper mentions residue-distance-based graph building but does not explain where RNA 3D coordinates come from (experimental, predicted, or derived from complexes). If coordinates come from bound complexes, structural leakage may occur.
Hotspot virtual node details missing. The “O-ring” hotspot-based virtual node is intriguing, but the paper doesn’t specify how hotspots are detected or whether this relies on complex-level data that could leak target information.
Knowledge distillation process underspecified. The paper does not specify which layers are distilled, the loss weighting between KD and regression, temperature parameters, or where supervision occurs in the student. These are crucial for reproducibility.
Training details incomplete. The description of model optimization, details on network structure and crucial dimensions for the protein, RNA, and fusion modules are missing. This makes it hard to reproduce.
Section redundancy. Sections 4.2 and 4.3 appear to describe similar cross-attention mechanisms, which creates confusion about what constitutes the final model.
Minor issues. The text has several typos (e.g., “distliation,” “attation”). Figure 1(c) appears to show the protein teacher outputting RNA embeddings—likely a labeling error. Table 4 is referenced in line 377 but should be Table 2.

**Questions:**

1.	What exact target variable is regressed (e.g., log Kd, Ka, ΔG) and in what units? Are affinity values normalized or transformed per dataset?
2.	How are RNA structural graphs derived for cases lacking 3D coordinates? Are predicted or unbound structures used?
3.	How are O-ring hotspots defined, and is this information available for unseen complexes during inference?
4.	What is the rationale for using the CoPRA (scratch) baseline instead of the full CoPRA variant which has better performance ?
Some comments:
1.	For reproducibility, a section that describing full architecture specifications for all  models (two teachers and the student, as well as fusion and prediction part), along with optimizer configurations, batch sizes, and training durations should be included in the paper.
2.	It would strengthen the paper to add results on at least one more benchmark dataset and include standard deviations or confidence intervals across folds. Most of the baseline results are taken from CoPRA, but in the original paper, there are more datasets and more benchmarks. I am wondering what the reason is that only one dataset and few baselines were chosen for this paper.
3.	The relevance of the teacher tasks to protein–RNA interaction learning could be discussed more clearly. In particular, it would be helpful to analyze whether the features learned from protein–protein and RNA–small molecule binding tasks truly generalize to protein–RNA interactions. What would happen if the same setup were used, but the teacher models did not rely on these specific interaction tasks and instead used generic protein and RNA encoders trained through self-supervised learning? Such an experiment would reveal whether the chosen pretraining tasks (PPI and RNA–small molecule) provide useful transferable representations for protein–RNA affinity prediction or if comparable performance could be achieved without them.

---

### Official Review · Reviewer_DQyd · 2025-11-02

**Soundness:** 3
**Presentation:** 2
**Contribution:** 3
**Rating:** 4
**Confidence:** 3

**Summary:**

This paper tackles two key challenges in protein–RNA binding affinity prediction: (1) the scarcity of experimentally measured protein–RNA complex data, which limits model training, and (2) insufficient cross-modal interaction modeling between proteins and RNAs. To address these issues, the authors propose KDPRA, a dual-molecule knowledge distillation framework with a bidirectional cross-attention fusion mechanism. Two separately trained teacher models for proteins and RNAs transfer structural and semantic knowledge to a student model, alleviating data scarcity through feature-level distillation. The bidirectional cross-attention module further captures residue-level protein–RNA interactions, aligning cross-modal features in a unified latent space. Additional components, such as structural hotspot priors and motif probing strategies, enhance interpretability. The experiment shows that this model outperforms other baseline models.

**Strengths:**

1. Innovative dual-teacher design: The proposed dual-teacher knowledge distillation effectively transfers prior knowledge from protein–protein and RNA–small molecule tasks, mitigating the problem of limited protein–RNA data.

2. Cross-modal interaction modeling: The bidirectional cross-attention module captures residue-level dependencies and achieves fine-grained alignment between protein and RNA features.

3. Comprehensive experiments: Extensive studies, including cross-validation, module and feature ablations, and contact distance sensitivity analyses, provide strong empirical support.

4. Interpretability and biological relevance: The motif probing module reveals key RNA binding motifs and regions, enhancing biological interpretability of the predictions.

**Weaknesses:**

1. Unclear model description: The logical flow among input, intermediate modules, and output is not well organized, making the model principle difficult to follow.

2. Limited transparency in knowledge distillation: The dual-teacher distillation mechanism lacks details on conflict resolution or feature space alignment, which may cause instability or redundancy.

3. Insufficient generalization analysis: Experiments are confined to the PRA_201 dataset without testing on unseen proteins/RNAs or cross-species scenarios.

4. Marginal or unstable module gains: Ablation results show limited performance drops after removing DSSP or RNA-FM, and unstable contribution from Virtual Nodes, suggesting potential feature redundancy or weak fusion.

**Questions:**

1. Could the authors elaborate on how the model handles potential conflicts or inconsistencies between the two teacher networks during distillation? Is there any mechanism to align their feature spaces or balance their influence on the student model?

2. Ablation results suggest that removing DSSP or RNA-FM causes only minor performance drops, and the Virtual Node module shows instability. Could the authors analyze whether certain features are redundant, and how the fusion process could be optimized?

3. Have the authors tested KDPRA on unseen proteins or RNAs, or on datasets from different organisms, to assess cross-domain or cross-species generalization?

4. Could the authors improve the clarity and logical flow of Section 4 (METHODS)? The current presentation of inputs, intermediate modules, and outputs is somewhat difficult to follow.

---

### Official Review · Reviewer_qVFS · 2025-11-03

**Soundness:** 2
**Presentation:** 2
**Contribution:** 2
**Rating:** 4
**Confidence:** 4

**Summary:**

This manuscript introduces KDPRA, a novel dual-molecule knowledge distillation framework for predicting protein–RNA binding affinity. The model employs a dual-teacher distillation mechanism (with one teacher model trained on protein–protein affinity prediction and another on RNA–small molecule affinity prediction) and integrates a bidirectional cross-attention fusion module to explicitly capture residue-level interactions between proteins and RNAs. On the PRA201 dataset, KDPRA achieves superior performance compared with baseline methods such as DeepNAP, FoldX, PredPRBA, and CoPRA. The authors also conduct extensive experiments, including module ablation, feature ablation, and a motif-probing analysis based on local perturbations, demonstrating the interpretability and biological relevance of the model.

**Strengths:**

The paper introduces a dual-teacher knowledge distillation framework; the model’s core bidirectional cross-attention mechanism can, in theory, capture fine-grained, residue-level interactions between proteins and RNA. In addition, by integrating multi-source features and designing a motif detection module, the study demonstrates thoughtful consideration in feature engineering and model interpretability.

**Weaknesses:**

The main current weaknesses of this manuscript lie in the rigor of methodological description, the depth of innovation justification, and the sufficiency of experimental validation. Specifically, the novelty of the dual-teacher knowledge distillation framework needs to be reinforced through clear comparisons with prior works (e.g., KIDA, FusionDTA) and key ablation studies, to demonstrate its unique value beyond mere data augmentation. The Methods section exhibits significant structural confusion and contradictory descriptions, requiring a reorganization to clarify the central role of bidirectional cross-attention and to precisely correct errors in feature dimensionality (especially regarding the virtual node). Furthermore, the rigor and insightfulness of the experiments need enhancement, which necessitates providing complete details on data processing and model construction to ensure reproducibility, as well as introducing cross-attention heatmap visualizations and systematic motif evaluations to offer intuitive biological evidence for model decisions. These improvements would elevate the paper from a performance report to a comprehensive study with both innovation and high credibility.

**Questions:**

1. The dual-teacher knowledge distillation (KD) framework has already been applied in other biomolecular prediction tasks (e.g., DTA, PPI), with similar implementations such as KIDA and FusionDTA. It is recommended to further emphasize the specific innovations of this work in the protein–RNA context. Firstly, in the Introduction, the authors should clearly state how their dual-teacher KD approach differs from existing work. If the method mainly transfers an existing KD framework to the protein–RNA binding prediction task, the originality claims should be appropriately moderated. In the Related Work section, a comparison table should be added to show differences between existing KD methods and KDPRA in terms of teacher task types, distillation levels, and loss functions, along with an explanation of why these existing approaches cannot be directly applied to protein–RNA binding prediction. In the Methods section, the authors need to specify at which level the distillation occurs and provide complete distillation loss formulas and hyperparameter definitions. In the Results section, key ablation and control experiments should be added to demonstrate the unique advantages of dual-teacher KD and rule out gains merely from multi-task learning or data augmentation.
2. Section 4.2 describes bidirectional cross-attention, including both protein-to-RNA and RNA-to-protein directions, using formulas (1)–(5). However, Section 4.3, titled “Cross-Attention Fusion Module,” actually only covers the protein-to-RNA direction. The authors claim in Section 4.2 that the “bidirectional cross-attention” is a major innovation of the model. The “cross-attention fusion module” in Section 4.3 overlaps functionally with 4.2 but does not clarify its relationship. Is it a basic version replaced by 4.2, or are two separate cross-attention modules used in the model? This relationship is not clearly explained.
3. The PRA201 dataset is derived from multiple sources (PDBbind, PRBABv2, ProNAB), but the paper does not clearly describe redundancy removal (e.g., how highly similar samples for the same protein or RNA are handled) nor the train/validation/test split strategy.
4. The study uses protein–protein and RNA–small molecule tasks as teachers, but these tasks may have different physicochemical characteristics from the target protein–RNA task. Directly using cosine similarity as the loss may ignore domain differences, potentially reducing knowledge transfer effectiveness. I recommend using t-SNE or UMAP for dimensionality reduction visualization to validate the efficacy of cross-task distillation.
5. In Section 5.1, the authors state: “...ESM-3B pre-trained model (2560 dimensions), combined with DSSP secondary structure features (14 dimensions) and an additional virtual node feature (2574 dimensions).” Based on this description, the final protein feature dimension seems to be: 2560 (ESM) + 14 (DSSP) + 2574 (Virtual Node) = 5148 dimensions. However, in the Supplementary Material, the total dimension is reported as 2574, which is clearly inconsistent. The authors may have mistakenly described this total as the “virtual node” dimension, or there may be an error in reporting the virtual node feature.
6. The authors mention encoding RNA structure using a GAT, but do not explain how the RNA graph topology is constructed, which directly affects reproducibility. The paper should provide detailed RNA graph construction methods, including precise definitions of nodes and edges, and describe strategies for handling missing experimental structures (e.g., using tools like RNAfold to predict secondary structures for graph construction).
7. Although the bidirectional cross-attention module is described in detail, no attention maps are shown. Visualizing attention maps or residue–residue contact heatmaps for protein–RNA pairs would greatly enhance the model’s interpretability and biological credibility.
8. Model interpretability should be systematic and generalizable to support reliable biological insights. The authors only present a single example (the 1MMS complex), showing that the fragment “UCAC” has the largest impact on model output. A more systematic evaluation is needed.

---

### Note · Authors · 2025-11-22

I have read and agree with the venue's withdrawal policy on behalf of myself and my co-authors.